# Associations between the Severity of Sarcopenia and Health-Related Quality of Life in Community-Dwelling Middle-Aged and Older Adults

**DOI:** 10.3390/ijerph18158026

**Published:** 2021-07-29

**Authors:** Raquel Fábrega-Cuadros, Fidel Hita-Contreras, Antonio Martínez-Amat, José Daniel Jiménez-García, Alexander Achalandabaso-Ochoa, Leyre Lavilla-Lerma, Patricia Alexandra García-Garro, Francisco Álvarez-Salvago, Agustín Aibar-Almazán

**Affiliations:** 1Department of Health Sciences, Faculty of Health Sciences, University of Jaén, 23071 Jaén, Spain; rfabrega@ujaen.es (R.F.-C.); fhita@ujaen.es (F.H.-C.); amamat@ujaen.es (A.M.-A.); aaochoa@ujaen.es (A.A.-O.); llavilla@ujaen.es (L.L.-L.); aaibar@ujaen.es (A.A.-A.); 2MOVE-IT Research Group, Department of Physical Education, Faculty of Education Sciences, University of Cádiz, 11003 Cádiz, Spain; 3GIP Pedagogy Research Group, Faculty of Distance and Virtual Education, Antonio José Camacho University Institution, Santiago de Cali 760001, Colombia; vidaymovimiento2012@hotmail.com; 4Department of Physiotherapy, Faculty of Health Sciences, European University of Valencia, 46112 Valencia, Spain; salvagofran@gmail.com

**Keywords:** sarcopenia, older adults, quality of life, anxiety, depression

## Abstract

(1) Background: The aim of this study was to analyze the associations between severity of sarcopenia and health-related quality of life (HRQoL) among community-dwelling middle-aged and older adults. (2) Methods: A cross-sectional study involving 304 older-adult participants was used to assess the severity of sarcopenia by measuring muscle strength (handgrip dynamometer), muscle mass (bioelectrical impedance analysis), and physical performance (Timed Up-and-Go test). The generic 36-item Short-Form Health Survey (SF-36) was used to evaluate HRQoL. Anxiety and depression (Hospital Anxiety and Depression Scale) as well as age were considered as possible confounders. Probable sarcopenia was determined by low muscle strength; confirmed sarcopenia was defined by the presence of both low muscle strength and muscle mass; and severe sarcopenia was defined by low muscle strength and mass along with poor physical performance. (3) Results: The linear regression analysis showed that the presence of probable sarcopenia was associated with the SF-36 domains physical role (adjusted R^2^ = 0.183), general health (adjusted R^2^ = 0.290), and social functioning (adjusted R^2^ = 0.299). As for the SF-36 mental (MCS) and physical (PCS) component summary scores, probable sarcopenia, as well as depression and anxiety, remained associated with MCS (adjusted R^2^ = 0.518), and these three variables, together with age, were linked to PCS (adjusted R^2^ = 0.340). (4) Conclusions: Probable sarcopenia, but not confirmed or severe sarcopenia, was independently associated with poor HRQoL. More precisely, it was related to PCS and MCS, as well as to the physical role, general health, and social functioning of SF-36 domains.

## 1. Introduction

Sarcopenia is defined as the generalized and progressive loss of skeletal muscle strength and mass, which can lead to adverse outcomes such as physical disability, reduced physical functions, and poor quality of life [1]. Although the first definition of sarcopenia elaborated by The European Working Group on Sarcopenia in Older People-1 (EWGSOP1) only considered muscle mass [2], the European Working Group on Sarcopenia in Older People-2 (EWGSOP2) more recently proposed three diagnostic parameters: low muscle strength, low muscle quality/quantity, and poor physical performance [3]. According to the definition, the prevalence of sarcopenia in older adults living in the community may vary between 1 and 50%, with higher rates with advanced age [4]. Furthermore, it is estimated that 5–13% of persons aged 60 to 70 years, and 11–50% aged 80 and over have sarcopenia [5].

Sarcopenia has been associated with multiple adverse physical effects (e.g., alterations in movement patterns, loss of independence and the need for long-term care [6]) and psychological effects, most commonly anxiety and depression. The prevalence of depression among the older population worldwide is around 7% and increases with age [7]. The incidence of anxiety in people over 60 is between 0.7 and 18.6%, with lower frequency among the elderly than in younger adults [8].

All the previous factors, together with sarcopenia, may contribute to a decrease in quality of life to the extent that it is conditioned by the physical and mental state and symptoms of elderly individuals [9]. Poor quality of life in this population has been associated with several negative health factors, including falls, admission to a nursing home, and mortality [10,11]. Many studies have checked the association between sarcopenia and quality of life, both physical and psychologically, in older people, even though it has not yet been studied in the Spanish population, for example: the study of Patel et al., reported a reduction in the quality of life in sarcopenic patients living in the United Kingdom [12], and another study presented a significantly higher number of mobility problems, habitual activity and anxiety in sarcopenic people [13]. Furthermore, the SarchoPhage Project, in a longitudinal study made up of 534 elderly, demonstrated that subjects who were diagnosed with sarcopenia were older and had poor quality of life and higher risk of falling [14]. Despite this, it seems necessary to carry out more studies to evaluate the real impact of sarcopenia on quality of life since it is a fundamental factor for older people [15].

For all this, one of the main goals of healthcare efforts concerning the elderly is the maintenance or improvement of their quality of life, a trend that has sparked widespread international interest and is supported by many action plans addressing the consequences of aging [16].

The objective of this study was to analyze the associations between the severity of sarcopenia and health-related quality of life (HRQL) in community-living older and middle-aged adults.

## 2. Materials and Methods

### 2.1. Study Design and Participants

This was a cross-sectional analytical study that involved a total of 304 elderly and middle-aged adults. It was approved by the Research Ethics Committee of the University of Jaén (TES 18/22 November) and carried out according to the Declaration of Helsinki, good clinical practices, and all applicable laws and regulations. Before the start of the study, all participants gave informed written consent.

To be included in the study, participants had to be community-based outpatient adults aged 50 years or older, have the ability to understand and complete the questionnaires, and signed an informed consent form. All subjects who suffered from limitations in physical activity, a chronic or severe illness, or any neuropsychiatric disorder that could influence their responses to the questionnaires were excluded.

### 2.2. Study Parameters and Definitions

#### 2.2.1. Definition

Sarcopenia was defined following the guidelines published in 2019 by the European Working Group on Sarcopenia in Older People (EWGSOP-2) [3]. Probable sarcopenia was determined by low muscle strength; confirmed sarcopenia was defined by the presence of both low muscle strength and muscle mass; and severe sarcopenia was diagnosed when low muscle strength and muscle mass appeared together with low physical performance.

#### 2.2.2. Muscle Strength

Muscle strength (kg) was determined by an analogue handgrip dynamometer (TKK 5001, Grip-A, Takei, Tokyo, Japan) following a previously described protocol [17]. Measurements were made twice on the dominant hand, and the mean value was taken into account [18]. A threshold of 16 kg for women and 27 kg for men was used to assess low muscle strength [19].

#### 2.2.3. Muscle Mass

Muscle mass was assessed by bioelectrical impedance analysis (BIA, InBody 720, Biospace Co. Ltd.; Seoul, Korea). Skeletal muscle mass (SM, kg) was calculated using the following BIA equation [20]: Skeletal muscle mass = [0.401 × (height^2^/resistance) + (3.825 × sex) − (0.071 × age) + 5.102]. Height was assessed in cm and resistance in ohms. Resistance depended on the water contained in the body, which has a constant proportion in muscle mass, since 73% of muscles are water. By relating this data to the age, sex and height of the individual, the muscle mass of the entire body could be calculated. The R represents the resistance of the tissues to the passage of an electric current. The resistance is proportional to the length of the body (generally its length or height is considered) and inversely proportional to the sectional area (generally the measurements that represent the perimeters of the trunk and limb segments) [21]. The Skeletal Muscle Mass Index (SMI) was obtained by dividing skeletal muscle mass by height in meters squared (kg/m^2^). To determine low muscle mass, thresholds of 6.42 kg/m^2^ (women) and 8.87 kg/m^2^ (men) for the SMI were used [22].

#### 2.2.4. Physical Performance

Physical performance was assessed by gait speed through the Timed Up-and-Go (TUG) test [1]. The resulting time (s) was converted into an estimate of gait speed by applying the formula (6/(TUG time)) × 1.62. Values ≤ 0.8 m/s were considered low gait speed [23].

#### 2.2.5. Anxiety and Depression

The Hospital Anxiety and Depression Scale (HADS) was used to assess the participants’ level of depression and anxiety [24,25]. This instrument is composed of 14 items, of which seven are related to depression and seven to anxiety. A high score on this questionnaire represents worse levels of depression and anxiety.

#### 2.2.6. Health-Related Quality of Life

The SF-36 questionnaire was also completed by all participants. It is a widely used questionnaire to assess generic health-related quality of life [26], and the Spanish version validated by Alonso et al. [27] was used. This tool has a total of 36 items classified into 8 scales: physical function, physical role, general health, vitality, bodily pain, social function, emotional role, mental health, and health change, as well as the physical and mental component summary scores. In this questionnaire, a score between 0 and 100 can be obtained, with the higher scores representing better quality of life.

### 2.3. Data Analysis

Continuous and categorical variables were expressed by means and standard deviations and by frequencies and percentages, respectively. Variable normality was assessed with the Kolmogorov–Smirnov test. To study possible individual differences in HRQoL as assessed by the SF-36 domains and component summaries scores (dependent variables) between groups according to the severity of sarcopenia (independent variables), a Student’s *t* test was used. Pearson’s correlation analysis was employed to evaluate associations of HRQoL with possible confounders (age, anxiety and depression). A stepwise linear regression model was used to determine independent associations. To evaluate the effect size coefficient of multiple determinations, adjusted R^2^ was used, which can be classified as follows: <0.02, insignificant; between 0.02 and 0.15, small; between 0.15 and 0.35, as medium; and >0.35, large [28]. Results were considered statistically significant for *p* < 0.05. Data analysis was carried out using the SPSS statistical package for social sciences for Windows (SPSS Inc., Chicago, IL, USA).

## 3. Results

In total, 304 participants (83.88% women) took part in this study. Table 1 shows the descriptive characteristics for the final sample. The mean HADS scores were 5.74 ± 4.02 (anxiety) and 4.99 ± 3.44 (depression). Concerning the SF-36, the PCS score was 68.61 ± 21.28 and the MCS score was 73.20 ± 20.35, while the higher and the lower domain scores were obtained by social functioning and physical functioning, respectively.

Table 2 displays the individual association between HRQoL and the severity of sarcopenia. Our findings showed that probable sarcopenia was associated with a worse HRQoL in all SF-36 domains except emotional role, general health, and vitality. However, fewer associations were observed with a confirmed diagnosis of sarcopenia (physical functioning, bodily pain, physical role, PCS, and MCS) or severe sarcopenia (social functioning and general health).

Regarding the confounding variables, the analysis of the bivariate correlation (Table 3) showed that higher levels of both anxiety and depression were associated with worse HRQoL in all SF-36 domains and component summaries. Nevertheless, older age was only related to poorer HRQoL in physical functioning, general health, vitality, and PCS. Regarding sex, women showed a better HRQoL in the SF-36 domains physical functioning, physical role, and mental health.

The linear regression analysis was employed to detect any variable that was independently associated with HRQoL (Table 4). As for the component summaries of the SF-36, probable sarcopenia, as well as anxiety and depression, remained associated with MCS (adjusted R^2^ = 0.523), and these three variables, together with sex and age, were independently related to PCS (adjusted R^2^ = 0.358). Depression and anxiety remained as independent predictors of a worse HRQoL in all domains except for vitality, where depression was the only associated factor (adjusted R^2^ = 0.231). In fact, these were the only factors associated with an emotional role (adjusted R^2^ = 0.212), and mental health (adjusted R^2^ = 0.582), and, along with higher age, were independently related to general health (adjusted R^2^ = 0.316). Together with other factors, the presence of probable sarcopenia was associated with the SF-36 domains physical functioning (adjusted R^2^ = 0.301), physical role (adjusted R^2^ = 0.186), pain (adjusted R^2^ = 0.191), and social functioning (adjusted R^2^ = 0.306).

## 4. Discussion

The purpose of this study was to analyze the association between sarcopenia and its levels of severity, anxiety, and depression on the one hand, and the health-related quality of life in older adults on the other. The main findings of the study showed that the probability of having sarcopenia is associated with a poorer quality of life. The SF-36 domains that displayed significant associations were PCS, MCS, physical role, general health, and social functioning. Probable sarcopenia was associated with HRQoL, which is a big step for primary care diagnosis, for example. Thus, from the first stage, the detection of sarcopenia can be carried out in a simple way, and could represent an important advance by establishing treatment early. Its increased severity would mean a worse quality of life, according to our results, but more studies are needed.

The percentages of probable, confirmed, and severe sarcopenia varied across the studies. Hu et al. [29] reported an 18.5% prevalence according to the criteria set by the Asian Working Group for Sarcopenia (AWGS). These low percentages are in agreement with results described by Kwon et al. [30], who reported a prevalence of 14.3% among the general population (18.7% among and 9.7% among women) using densitometry for an average age of 44.1 ± 0.2. In the study by Ida et al. [31] the prevalence of sarcopenia as measured by the SARC-F questionnaire was 22.5%. Locquet et al. [32], employing six existing definitions, reported values between 5.9 and 32.5%. Öztürk et al. [33] described a prevalence value of 14%, in agreement with a similar study that set prevalence at 13.30% [34]. Such differences are due to ample variations with a wide range of causes, among which are the disparate definitions and diagnostic measuring instruments [35]. In this sense, there are numerous investigations that relate grip strength with quality of life [36,37,38] and other factors but not with the severity or degrees of sarcopenia.

As far as mood alterations are concerned, the results were similar to those described in a study carried out in Spain that analyzed two different cohorts of community-living groups over 60 years of age (2376 and 1911 participants) [39]. In comparison with other age groups, mental health and anxiety disorders have received scant attention in the literature, despite the fact that anxiety has been described as the silent geriatric giant for its prevalence among older adults [40]. A recent systematic review and meta-analysis [41] that included eight studies carried out in Spain reported an 11% overall prevalence for the over-65 population. As for depression, a recent prevalence study [7] among middle-aged and older adults revealed that, after a four-year follow-up, overall incidence was 22.3%, with much higher percentages in rural areas (25.7%) and among women (27.9%). Compared with individuals who described their health status as poor, those who reported excellent health were 62% less likely to develop depressive symptoms. As in the case of anxiety, depression is a serious mental disorders among older adults, with prevalence values ranging from 1 to 16%, and between 7.2 and 49% of older adults presenting symptoms of depression, according to a systematic review. Furthermore, symptoms of depression and anxiety have been linked to increased risk of disability towards the end of life, which is all the more reason to improve it diagnosis and prevention [42].

In this study, an association was found between the diagnosis of sarcopenia and the domains of physical function, the role-physical, pain, PCS, and MCS. Severe sarcopenia was associated with general health and social function. In our case, probable sarcopenia was associated with poorer quality of life except for the domains of general health, vitality, and emotional role. When a regression analysis was performed, the association with the role-physical, general health, and social function domains of SF-36 was maintained. Other authors have reported similar results, such as Kull et al. [43], who found a reduction of the quality of life in the SF-36 domains of physical function and vitality among sarcopenia patients. These individuals also scored markedly lower on the physical role, emotional role, and vitality subscales of the SF-36 questionnaire. In the case of Patel et al. [12], they were able to confirm the affected domains as general health and physical function, and that both sarcopenic women and men scored low on the self-reported general health and functional domains. The results described in the scientific literature are non-conclusive regarding the associations between sarcopenia, its levels of severity and quality of life although many studies prove this association. In Beaudart et al. [14] sarcopenic subjects were found to have poorer quality of life in relation to physical health for the physical function domain. Silva Neto et al. [38] wrote that sarcopenia was not associated with quality of life in women after an evaluation of body composition with BMI and densitometry, grip strength with a dynamometer, and quality of life with the SF-36 health questionnaire.

The mean age of the participants was 64.92 ± 5.74 years and, despite the fact that no statistically significant differences were found between quality of life and the variables studied among the participants suffering from sarcopenia or sarcopenic obesity, the values were lower in those affected by the condition. These results were similar to those of Marques et al. [36], who used densitometry for the calculation of body composition, and in which sarcopenia was negatively associated with the quality of life of men, which agreed with our results.

As for anxiety and depression, our study revealed that they were associated with a worse HRQoL in all SF-36 domains and component summaries. Depression and anxiety remained independent predictors of poor HRQoL in all domains except for the emotional role, where depression was the only associated factor. Furthermore, depression and anxiety were the only variables independently associated to pain and vitality. With similar results, a longitudinal study [44] looking into how anxiety and depression influence individual quality of life in old age involved a Portuguese population over 50 who participated in cycle 4 (S4) and cycle 6 (S6) of the project known as the European Health, Aging, and Retirement Survey (SHARE)––1765 participants in cycle 4 (study start), and 1201 in cycle 6 (follow-up). All participants were evaluated using the CASP-12 scale, which measures quality of life, and the Euro-D scale, which measures emotional state and is composed of five items from the anxiety-measuring Beck inventory. To identify the factors associated with changes in quality of life, linear models of mixed effects were carried out. Their results showed an association between increased age and poor quality of life, which, in turn, was associated with higher levels of anxiety and depression and low education. Out of the variations measured by CASP-12, 42.1% may be due to random and fixed effects during aging, considering depression and anxiety as the factors with the largest relative importance. These two factors play a fundamental role in the quality of life of the elderly and must be recognized as primary intervention domains for the promotion of active and healthy aging. These results, coming from a long-term study, are very important although we must acknowledge that they are not comparable due to different measuring tools having been used (in this case the SF-36). In a recent 2019 study [45] based on the Korean National Health and Nutrition Survey, people with sarcopenia (assessed with appendicular muscle mass index) had greater anxiety and depression than non-sarcopenic people, and Chen et al. [46] showed that depressive symptoms increase in chronic conditions such as sarcopenia. To establish valid comparisons, further studies will be required that apply the same methodology.

Older age only appeared to be related to a worse HRQoL in physical functioning, general health, vitality, and PCS. In the regression analysis, age was associated with physical functioning, mental health, and general health. In the study by Cho et al. [47], sarcopenia had a significant association with perceived stress and suicidal ideation, and had a negative association with quality-of-life indicators although these patterns were more significant in people below 60. Sarcopenia was associated with poor mental health and decreased quality of life, but their sample was younger than ours. Another study concerning age concluded that, at the three-year follow-up, the summary score of the reduced physical component of SF-36 was independently associated with a reduced SPPB score and an increase in walking time for the 400 m distance [48].

Some sex differences were revealed by our study, as women had a better HRQoL in the physical role, physical function, and mental health domains of the SF-36 questionnaire and sex was revealed to be related to mental and general health. Silva Neto et al. [38] found, in a sample of 56 older women, significant associations between variables related to quality of life (measured with the SF-36) and sarcopenia, as well as muscle mass and sarcopenic obesity. Specifically, 13 women (23.21%) were classified as sarcopenic while 76.78% (43) were not. These results are similar to those described by Marques et al. [36], who used densitometry to calculate body composition, and in which sarcopenia was negatively associated with quality of life in men. Go et al. [13] showed that individuals suffering from sarcopenia presented increased difficulties in mobility, usual activities and self-care, and also had higher levels of anxiety. The same study looked into the associations between sarcopenia and quality of life in a sample of men from Korea. Their results showed that sarcopenia had a greater impact on those dimensions corresponding to physical functioning than on mental health and social functioning. Although only in men did BMI (body mass index), single leg stance time, leg press power, leg press strength, SF-36 questionnaire general health score, total physical performance test score, and bioavailable testosterone levels correlate with sarcopenia [49]. Studies involving both sexes and those studying differences in individuals are needed. Finally, we found that the component summaries of SF-36, depression, anxiety, and probable sarcopenia remained associated with MCS, and these three variables, together with age, were independently related to PCS.

Our study suffered from some limitations that should be taken into consideration. Densitometry could not be used to evaluate the muscle mass although BIA has been recommended for this purpose [3]. Furthermore, although the strength and muscle mass of all participants were measured, the SARC-F questionnaire was not used (although it had been recently validated for the Spanish population [50] and is recommended by the EWGSOP2 as an appropriate sarcopenia screening method). This study was performed on older adults, and most of the participants were women; thus, any generalization of its results should be limited to individuals of similar characteristics. The presence of comorbidities or other potential confounders (household income or education) that may influence the association between sarcopenia and quality of life were not controlled, and future studies should take this into account. Finally, we could not evaluate the sensitivity to change due to the transversal nature of the study. A longitudinal design would be required to evaluate this aspect, a design that should be considered by future studies.

## 5. Conclusions

The findings of this study suggest that, in community-dwelling middle-aged and older adults, probable sarcopenia, but not confirmed or severe sarcopenia, was independently associated with a poor HRQoL. Our results showed that probable sarcopenia, anxiety and depression were associated with the MCS summary component, and these three, together with sex and age, were independently associated with PCS. Probable sarcopenia was also associated with physical functioning, physical role, pain, and social functioning. Depression and anxiety were independent predictors of a worse HRQoL in all domains except for vitality, where depression was the only associated factor. They were also associated with emotional role, and mental health and, along with higher age, were independently related to general health. These findings allowed us to suggest that, considering its effects on quality of life, diagnoses of probable sarcopenia (its first degree) should be considered as an important warning regarding programs for the prevention and treatment of this condition among community-dwelling middle-aged and older adults.

## Figures and Tables

**Table 1 ijerph-18-08026-t001:** Descriptive characteristics of the participants (*n* = 304).

Characteristics		Mean	SD	Frequency	Percentage
Age (years)		72.04	7.88		
Sex	Women			255	83.88
	Men			49	16.12
Occupational status	Not working			271	89.14
	Working			33	10.86
Academic level	Primary or less			188	61.84
	Secondary or higher			116	38.16
HADS	Anxiety	5.74	4.02		
	Depression	4.99	3.44		
SF-36	Physical functioning	64.18	25.18		
	Physical role	67.68	35.62		
	General health	64.96	21.11		
	Bodily pain	69.91	29.12		
	Vitality	70.33	26.83		
	Social functioning	82.34	21.74		
	Emotional role	74.40	33.99		
	Mental health	67.76	19.93		
	PCS	68.61	21.28		
	MCS	73.20	20.35		

HADS: Hospital Anxiety and Depression Scale. MCS: Mental component summary. PCS: Physical component summary. SD: Standard deviation. SF-36: 36-item Short-Form Health Survey.

**Table 2 ijerph-18-08026-t002:** Health-related quality of life according to the severity of sarcopenia (*n* = 304).

Outcomes	Probable Sarcopenia					Confirmed Sarcopenia					Severe Sarcopenia				
SF-36	No (*n* = 172)		Yes (*n* = 132)		*p*-value	No (*n* = 232)		Yes (*n* = 72)		*p*-value	No (*n* = 291)		Yes (*n* = 13)		*p*-value
	Mean	SD	Mean	SD		Mean	SD	Mean	SD		Mean	SD	Mean	SD	
Physical functioning	69.01	23.86	57.89	25.54	<0.001	65.82	24.88	58.90	25.58	0.042	64.76	25.18	51.15	22.00	0.056
Physical role	75.29	31.53	57.77	38.23	<0.001	70.26	34.49	59.38	38.10	0.023	67.87	35.40	63.46	41.60	0.663
General health	66.94	20.88	62.39	21.21	0.062	66.27	20.92	60.76	21.31	0.053	65.49	20.77	53.08	25.86	0.038
Bodily pain	74.49	25.65	63.94	32.24	0.002	72.00	27.90	63.16	32.03	0.024	70.20	29.18	63.46	28.05	0.415
Vitality	72.36	25.60	67.69	28.25	0.133	71.60	25.99	66.26	29.21	0.140	70.90	26.75	57.65	26.51	0.082
Social functioning	85.69	19.00	77.97	24.24	0.003	83.40	20.97	78.90	23.87	0.153	82.88	21.48	70.23	24.75	0.040
Emotional role	76.93	32.31	71.10	35.91	0.138	75.60	33.26	70.53	36.19	0.269	74.58	33.82	70.49	38.81	0.672
Mental health	70.08	19.04	64.73	20.71	0.020	68.72	19.17	64.65	22.05	0.131	67.89	19.72	64.77	24.92	0.582
PCS	73.15	18.57	62.69	23.14	<0.001	70.52	20.44	62.47	22.87	0.005	69.07	21.11	58.38	23.39	0.077
MCS	76.26	19.17	69.21	21.20	0.003	74.50	19.76	69.01	21.76	0.045	73.53	20.18	65.80	23.53	0.181

MCS: Mental component summary. PCS: Physical component summary. SD: Standard deviation. SF-36: 36-item Short-Form Health Survey.

**Table 3 ijerph-18-08026-t003:** Associations between HRQoL and anxiety, depression, and age (*n* = 304).

Outcomes	Anxiety		Depression		Age	
SF-36	r	*p*-value	r	*p*-value	r	*p*-value
Physical functioning	−0.464	<0.001	−0.405	<0.001	−0.184	0.001
Physical role	−0.316	<0.001	−0.307	<0.001	−0.043	0.452
General health	−0.400	<0.001	−0.471	<0.001	−0.329	<0.001
Bodily pain	−0.363	<0.001	−0.388	<0.001	−0.105	0.067
Vitality	−0.364	<0.001	−0.480	< 0.001	−0.134	0.020
Social functioning	−0.488	<0.001	−0.490	<0.001	−0.107	0.062
Emotional role	−0.432	<0.001	−0.395	<0.001	0.008	0.893
Mentalhealth	−0.727	<0.001	−0.638	<0.001	0.016	0.784
PCS	−0.466	<0.001	−0.496	<0.001	−0.247	<0.001
MCS	−0.656	<0.001	−0.639	<0.001	−0.062	0.278

HRQoL: Health-related quality of life. MCS: Mental component summary. PCS: Physical component summary. r: Pearson’s correlation coefficient. SF-36: 36-item Short-Form Health Survey.

**Table 4 ijerph-18-08026-t004:** Multivariate linear regression analyses for variables associated with HRQoL (*n* = 304).

SF-36	Variables	B	β	95% CI		*p*-Value
Physical functioning	Anxiety	−20.254	−0.360	−30.077	−10.432	<0.001
	Age	−0.550	−0.172	−0.878	−0.222	0.001
	Probable sarcopenia	−7.294	−0.144	−12.240	−2.348	0.004
	Sex	−8.418	−0.123	−15.270	−1.566	0.016
	Depression	−0.980	−0.134	−1.954	−0.007	0.048
Physical role	Anxiety	−1.307	−0.147	−2.529	−0.085	0.036
	Probable sarcopenia	−14.938	−0.208	−22.370	−7.506	<0.001
	Sex	−15.831	−0.164	−26.220	−5.442	0.003
	Depression	−2.086	−0.201	−3.510	−0.662	0.004
General health	Depression	−1.478	−0.241	−2.262	−0.693	<0.001
	Age	−0.786	−0.293	−1.052	−0.520	<0.001
	Anxiety	−1.398	−0.266	−2.055	−0.740	<0.001
Bodily pain	Depression	−2.077	−0.245	−3.199	−0.955	<0.001
	Anxiety	−1.441	−0.199	−2.396	−0.487	0.003
	Probable sarcopenia	−7.738	−0.132	−13.787	−1.689	0.012
Vitality	Depression	−3.749	−0.480	−4.524	−2.974	<0.001
Social functioning	Depression	−1.807	−0.286	−2.583	−1.032	<0.001
	Anxiety	−1.616	−0.299	−2.276	−0.956	<0.001
	Probable sarcopenia	−5.048	−0.115	−9.231	−0.866	0.018
Emotional role	Anxiety	−2.571	−0.304	−3.669	−1.474	<0.001
	Depression	−2.012	−0.204	−3.294	−0.730	0.002
Mental Health	Anxiety	−2.671	−0.538	−3.140	−2.202	<0.001
	Depression	−1.730	−0.299	−2.278	−1.182	<0.001
PCS	Depression	−1.624	−0.262	−2.413	−0.835	<0.001
	Anxiety	−1.513	−0.286	−2.180	−0.846	<0.001
	Age	−0.534	−0.198	−0.800	−0.268	<0.001
	Probable sarcopenia	−6.505	−0.152	−10.514	−2.495	0.002
	Sex	−5.722	−0.099	−11.277	−0.167	0.044
MCS	Anxiety	−2.134	−0.421	−2.646	−1.622	<0.001
	Depression	−2.134	−0.361	−2.736	−1.531	<0.001
	Probable sarcopenia	−3.771	−0.092	−7.019	−0.524	0.023

B: Unstandardized coefficient. β: Standardized coefficient. CI: Confidence interval. HRQoL: Health-related quality of life. MCS: Mental component summary. PCS: Physical component summary. SF-36: 36-item Short-Form Health Survey.

## Data Availability

The data shown in this study are available upon request from the corresponding author. Data are not available to the public given the sensitive nature of the questions asked in this study and the necessary guarantees of privacy and confidentiality.

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
