# Peer review of "Associations between the Severity of Sarcopenia and Health-Related Quality of Life in Community-Dwelling Middle-Aged and Older Adults"

_ijerph, 2021, doi:10.3390/ijerph18158026_

Round 1

Reviewer 1 Report

This cross-sectional study in an older Spanish adult population addresses the association of sarcopenia and HRQoL.  There is prior evidence for such an association and, intuitively, this is not surprising.  This does not mean that a new study on this subject has nothing to add.  However, the paper needs to set out more clearly, the state of the art: what is the current evidence for and against such an association.  This needs to be summarised briefly in the introduction to provide a clear context and rationale for another study on this subject.  A more in-depth, but also synthetic, context should be provided in the discussion, so that the reader can clearly understand how this study has improved understanding of the field and what the clinical significance is.

Concerning the results, the finding that only first degree of sarcopenia was independently associated with HRQoL bears further discussion and explanation.  The fact that this did not appear to be the major driver of HRQoL in the multivariate analysis may also undermine the argument for its importance.  If it is a significant driver of HRQoL, why do the authors think that increasing severity of sarcopenia could not be shown to have increasing effect on HRQoL?

The impact of age is evident, but probably understated in this cohort due to the population only of older adults.  The study population was also heavily biased towards women, which is a limitation on its generalisability.

Author Response

This cross-sectional study in an older Spanish adult population addresses the association of sarcopenia and HRQoL.  There is prior evidence for such an association and, intuitively, this is not surprising.  This does not mean that a new study on this subject has nothing to add.  However, the paper needs to set out more clearly, the state of the art: what is the current evidence for and against such an association.  This needs to be summarised briefly in the introduction to provide a clear context and rationale for another study on this subject.  A more in-depth, but also synthetic, context should be provided in the discussion, so that the reader can clearly understand how this study has improved understanding of the field and what the clinical significance is.

Thank you for your comment. Modifications are in the manuscript

Concerning the results, the finding that only first degree of sarcopenia was independently associated with HRQoL bears further discussion and explanation.  The fact that this did not appear to be the major driver of HRQoL in the multivariate analysis may also undermine the argument for its importance.  If it is a significant driver of HRQoL, why do the authors think that increasing severity of sarcopenia could not be shown to have increasing effect on HRQoL?

Thank you for your comment. Modifications are in the manuscript.

The impact of age is evident, but probably understated in this cohort due to the population only of older adults.  The study population was also heavily biased towards women, which is a limitation on its generalisability.

We thank the reviewer with these comments. The limitation section has been modified.

Reviewer 2 Report

MDPI – IJERPH

This manuscript addresses the association between sarcopenia severity and health-related quality of life among community-dwelling middle-aged and older adults. The manuscript is addressing an interesting topic. However, there are major problems in statistical analysis. I think authors should revise their analyses before a closer evaluation of the paper.

Selecting only statistically significant variables from bivariate analysis, stepwise regression strategy and interpreting multiple adjusted effect estimates from a single model (Table 2 fallacy) are problematic practices. A fundamental problem with stepwise regression is that some real explanatory variables (like gender and education) that have causal effects on the dependent variable may happen to not be statistically significant and therefore not included in the final regression models.

Check the following articles:

https://journalofbigdata.springeropen.com/articles/10.1186/s40537-018-0143-6

https://www.sciencedirect.com/science/article/abs/pii/089543569600025X

https://academic.oup.com/aje/article/177/4/292/147738

Instead, authors should first draw assumptions about causal relationships of variables using directed acyclic graphs and then select appropriate variables for regression models. If the authors are interested in many predictors, they should draw and model each of them separately with directed acyclic graphs, and not Interpret coefficients from a single model. Guidance can be found for instance in these sources

https://bmcmedresmethodol.biomedcentral.com/articles/10.1186/1471-2288-8-70

https://academic.oup.com/ije/advance-article/doi/10.1093/ije/dyaa213/6012812

Furthermore, there is an important covariate that was totally missed in this study, Comorbidities. If the authors have information about chronic diseases and medical conditions. They should consider adding this variable as a potential confounder of the association between sarcopenia and health-related quality of life.

Minor points:

The manuscript has some grammar errors that are affecting the flow of the work.

Abstract:

There is no information on how sarcopenia severity was defined.

Introduction:

L 47:  Do you mean higher rates with advanced age?

L49: Reference (5) does not seem to be from the WHO as the test suggests.

L 52 -56: these two sentences seem to be out of context. Why discussing anxiety and depression in particular?

The rationale of the study should be improved. There is nothing regarding previous studies that assessed the association between sarcopenia and health-related quality of life.

Methods:

L 90: it would be useful to provide more information. How many trials? Which hand? What value was used, mean of scores, maximum score?

L96: what do you mean by resistance?

Author Response

This manuscript addresses the association between sarcopenia severity and health-related quality of life among community-dwelling middle-aged and older adults. The manuscript is addressing an interesting topic. However, there are major problems in statistical analysis. I think authors should revise their analyses before a closer evaluation of the paper.

Selecting only statistically significant variables from bivariate analysis, stepwise regression strategy and interpreting multiple adjusted effect estimates from a single model (Table 2 fallacy) are problematic practices. A fundamental problem with stepwise regression is that some real explanatory variables (like gender and education) that have causal effects on the dependent variable may happen to not be statistically significant and therefore not included in the final regression models.

Check the following articles:

https://journalofbigdata.springeropen.com/articles/10.1186/s40537-018-0143-6

https://www.sciencedirect.com/science/article/abs/pii/089543569600025X

https://academic.oup.com/aje/article/177/4/292/147738

Instead, authors should first draw assumptions about causal relationships of variables using directed acyclic graphs and then select appropriate variables for regression models. If the authors are interested in many predictors, they should draw and model each of them separately with directed acyclic graphs, and not Interpret coefficients from a single model. Guidance can be found for instance in these sources

https://bmcmedresmethodol.biomedcentral.com/articles/10.1186/1471-2288-8-70

https://academic.oup.com/ije/advance-article/doi/10.1093/ije/dyaa213/6012812

Furthermore, there is an important covariate that was totally missed in this study, Comorbidities. If the authors have information about chronic diseases and medical conditions. They should consider adding this variable as a potential confounder of the association between sarcopenia and health-related quality of life.

We thank the reviewer for these comments. In order to follow them, a linear regression analysis that included all the variables studied has been performed. Regarding other confouders, the goal of this study was to analyze the associations between the severity of sarcopenia and health-related quality of life, thus only some possible confounders were taken into account (age, anxiety and depression). We agree that comorbidities are an important covariate, but, in our opinion, they have been taken into account in the exclusion criteria (“limitations in physical activity, a chronic or severe illness, or any neuropsychiatric disorder…”).

Minor points:

The manuscript has some grammar errors that are affecting the flow of the work.

Thanks for the comment. Grammar errors have been resolved.

Abstract:

There is no information on how sarcopenia severity was defined.

Thanks for your appreciation. This has been modified in the abstract.

Introduction:

L 47:  Do you mean higher rates with advanced age?

Thank you for your comment. By that expression it was meant that there are higher rates with advanced age. It has been changed in the manuscript for your better understanding.

L49: Reference (5) does not seem to be from the WHO as the test suggests.

Thank you for your comment. It has been modified in the manuscript.

L 52 -56: these two sentences seem to be out of context. Why discussing anxiety and depression in particular?

Thank you very much for this comment. It has been modified in the manuscript.

The rationale of the study should be improved. There is nothing regarding previous studies that assessed the association between sarcopenia and health-related quality of life.

We agree with the reviewer's comment and the introductory section has been modified accordingly.

Methods:

L 90: it would be useful to provide more information. How many trials? Which hand? What value was used, mean of scores, maximum score?

Thank you for your comment. Measurements were made twice on the dominant hand, and the mean value was taken into account. It has been changed in the manuscript for your better understanding.

L96: what do you mean by resistance?

Thank you for your comment. Resistance depends on the water contained in the body, which has a constant proportion in muscle mass, since 73% of muscles are water. Taking this data and relating it to others such as age, sex and height of the individual, the muscle mass of the entire body can be calculated. The R represents the resistance of the tissues to the passage of an electric current. The reactance is due to the electrical effect of the charge offered for short periods, by the lipid component of the membranes of the cell mass. The resistance is proportional to the length of the body (generally its length or height is considered) and inversely proportional to the sectional area (generally the measurements that represent the perimeters of the trunk and limb segments). It has been changed in the manuscript for your better understanding.

Round 2

Reviewer 2 Report

I thank the authors for incorporating my comments within the revised version of manuscript. In general, the manuscript has improved. However, regarding the first point I raised, I think the authors should address not accounting for comorbidities and (other potential confounders of the association between sarcopenia and quality of life) in the limitations of the study.

Author Response

Thank you very much for your comments. The limitation section has been modified